# Fabrication of a T-Shaped Microfluidic Channel Using a Consumer Laser Cutter and Application to Monodisperse Microdroplet Formation

**DOI:** 10.3390/mi12020160

**Published:** 2021-02-05

**Authors:** Naoki Sasaki, Eisuke Sugenami

**Affiliations:** 1Department of Applied Chemistry, Faculty of Science and Engineering, Toyo University, 2100 Kujirai, Kawagoe, Saitama 350-8585, Japan; nsasaki2020@gmail.com; 2Department of Chemistry, College of Science, Rikkyo University, 3-34-1 Nishi-Ikebukuro, Toshima, Tokyo 171-8501, Japan

**Keywords:** microdroplets, microfluidics, monodisperse, artificial cell

## Abstract

The use of micrometer-sized droplets for chemical and biochemical analysis has been widely explored. Photolithography is mainly used to fabricate microfluidic devices, which is often employed to form monodisperse microdroplets. Although photolithography enables precise microfabrication, it is not readily available to biochemists because it requires specialized equipment such as clean room and mask aligners, and expensive consumables such as photoresist and silicon wafers. In this study, we fabricated a microfluidic device using a consumer laser cutter and applied it to droplet formation. Monodisperse microdroplets were formed by using an oil phase for droplet digital polymerase chain reaction (PCR) as the continuous phase and phosphate-buffered saline or polyethylene glycol solution as the dispersed phase. The droplet size decreased as the flow rate of the continuous phase increased and approached a constant value. The method developed in this study can be used to realize microdroplet-based biochemical analysis with simple devices or to construct artificial cells.

## 1. Introduction

Recently, many chemical and biochemical analyses using microdroplets have been reported [1,2,3,4]. A microdroplet is a small droplet with a diameter of from one to several hundred micrometers. By encapsulating biomolecules such as DNA and enzymes in these microdroplets, each droplet can be used as a reaction vessel. The typical application of microdroplets is droplet digital polymerase chain reaction (ddPCR) [5], in which a sample solution of unknown concentration is diluted with reagents and divided into a thousand of microdroplets, and each droplet contains one molecule of the target DNA or none at all. When polymerase chain reaction (PCR) is performed in this situation, DNA is amplified only in the droplet containing the target DNA, and fluorescence is observed when the fluorescent dye binds to the amplified DNA. Therefore, the number of droplets where fluorescence is observed corresponds to the number of target DNA molecules contained in the original sample, which enables absolute quantification without the need for a calibration curve. Therefore, the microdroplet is a useful system for the analysis of DNA and other biomolecules.

Another application of microdroplets is to create artificial cells [6,7] by encapsulating biomolecules inside microdroplets, which can be used to understand the structure and function of cells. In such microdroplets, it has been reported that the rate of GFP synthesis depends on the droplet size [8], and that the phase separation rate of a mixture of DNA and polyethylene glycol (PEG) depends on the droplet size [9]. Therefore, microdroplets are an extremely useful and promising system for clarifying the science of biochemical analysis.

Various techniques have been used to form microdroplets. Typical methods include vortexing [9], the use of porous structures [10], and the use of porous membranes [11]. However, it is difficult to fabricate monodisperse microdroplets with these methods. On the other hand, monodisperse microdroplets can be formed by using microfluidic devices [12,13,14]. This is a method to form droplets by dividing the dispersed phase by the continuous phase using micrometer-scale channels. Monodisperse microdroplets can be obtained by pumping the solutions at a constant flow rate using a syringe pump to maintain the shear force acting on the dispersed phase at a constant level.

Photolithography (including soft lithography based on the mold fabricated by photolithography) is one of the main fabrication methods for microfluidic devices [15,16]. However, although this technique enables accurate microfabrication, it requires expensive and extensive facilities and equipment such as clean rooms and mask aligners, expensive consumables such as photoresist and silicon wafers, and specialized software such as CAD. This is not something that biochemists who do not specialize in microfabrication can utilize. Recently, many methods for fabricating microfluidic devices other than photolithography have been reported [17]. A microfluidic device for microdroplet formation has also been reported [18], but it is not an easy method for biochemists to use because an industrial laser is used to fabricate the mold for the device.

We have developed an inexpensive and simple method for fabricating microfluidic devices using a consumer-grade CO_2_ laser cutter [19,20,21]. In this method, an acrylic sheet is cut out using a laser cutter and bonded to another acrylic sheet to make a mold, and then polydimethylsiloxane (PDMS) is poured into the mold to make a device with the same channel pattern as the mold. Compared to the conventional method using photolithography, this method is inexpensive, does not require expertise, and does not require CAD but employs a standard vector graphics editor (Adobe illustrator^®^) to design channel patterns. As described by Qin et al. [16], it takes a few hours to become familiar with Adobe illustrator^®^, while it may require a few weeks to learn the basics of CAD. Of course, various laser-based microfabrication methods have been reported, but, as reviewed recently in this journal by Puryear III [22], it is important for beginners to utilize the fabrication process with high familiarity. Unfortunately, current laser-based microfabrication methods remain complicated and often require expensive machines [22]. Considering the aforementioned characteristics of laser-based microfabrication methods, it is clear that biochemists who do not specialize in microfabrication can easily use our method, and that the first biochemist-friendly microfluidic device for microdroplet formation can be created using our method.

In this study, a T-shaped microfluidic channel is fabricated using a consumer-grade laser cutter and applied to the formation of monodisperse microdroplets. The concept is shown in Figure 1, where the aqueous phase is pumped from the bottom of the T-shaped microchannel and the oil phase is pumped from the left side to form water-in-oil type microdroplets. In addition to phosphate buffer saline (PBS), a concentrated aqueous solution of PEG is used as the aqueous phase, demonstrating that this method can be applied to the formation of microdroplets containing high concentrations of macromolecules as in cells.

## 2. Materials and Methods

Schematic illustrations of the procedure of fabricating the microfluidic device are shown in Figure 2. The device was fabricated as reported previously [19,20,21], with some modification to the experimental procedures. An acrylic plate was cut out of an acrylic sheet (0.2 mm thickness, CLAREX, Nitto Jushi Kogyo, Tokyo, Japan) with a laser cutter (HAJIME, Oh-Laser, Saitama, Japan) (Figure 2A). Based on the results of preliminary studies, the power of the laser was set to 8% of the full power and the sweep speed was set to 6 mm s^−1^. The acrylic plate was glued to another acrylic plate (1 mm thickness, CLAREX, Nitto Jushi Kogyo) with a glue (Acrysunday 14-3201, Acrysunday, Tokyo, Japan) to form a mold (Figure 2B). Poly(dimethylsiloxane) (PDMS) substrates were fabricated by pouring a prepolymer of PDMS (SILPOT 184, Dow Corning, Midland, MI, USA) on the mold. The prepolymer was cured in an oven at 65 °C for 60 min, and the cured PDMS was peeled off from the mold, bonded to a virgin glass slide, and cured again at 100 °C for 60 min (Figure 2C). Through-holes were punched at the end of the microchannel patterns on the PDMS substrate using metallic eyelets. Silicone tubes (0.5 mm id, 0.8 mm od, Taiyo Kogyo, Tokyo, Japan) were glued to the through-holes with the prepolymer at 100 °C for 60 min, and the PDMS substrate was bonded to another virgin glass slide (Figure 2D). In a separate experiment, the PDMS substrate was cut by a razor, and cross-sectional images of the channel patterns were obtained under an inverted microscope (IX71, Olympus, Tokyo, Japan) equipped with a CMOS camera (ORCA-Flash 4.0 V2, Hamamatsu Photonics, Hamamatsu, Japan) and a 4× objective lens. HCImage Software (Hamamatsu Photonics) was used to process the images.

PBS (T900, Takara bio, Shiga, Japan) or an aqueous solution of PEG (average MW: 7000–9000, 10%) were used as the dispersed phase. The dispersed phase was infused into the channel at 1 μL min^−1^ by a syringe pump (Legato 111, KD Scientific, Holliston, MA, USA). An oil for droplet digital PCR (1864005, Bio-Rad, Hercules, CA, USA) was used as the continuous phase. The continuous phase was infused into the channel at 4~40 μL min^−1^ by another syringe pump. Bright field images were acquired using an inverted microscope (IX71, Olympus) equipped with a CMOS camera (ORCA-Flash 4.0 V2, Hamamatsu Photonics) and a 4× objective lens. HSR Software (Hamamatsu Photonics) and Image J (National Institutes of Health) were used to process the images.

## 3. Results

### 3.1. Microfluidic Device

A picture of the microfluidic device is shown in Figure 3A. A T-shaped microfluidic channel was successfully fabricated without channel clogging. Microscopic images of cross-section of recessed microchannel patterns on the PDMS substrate are shown in Figure 3B–D. It can be seen that the channel pattern was formed roughly according to the size shown in Figure 1. As in our previous study [19], the cross-sectional shape of the channel was trapezoidal. The microfluidic channel was employed to form a plug flow (Figure 3E). In this study, a microfluidic device consisting of a PDMS substrate and a glass substrate reversibly bonded to each other was used, and the flow rate of the continuous phase was set to a maximum of 40 μL min^−1^, but there was no sign of leakage of the solution out of the flow path.

### 3.2. Microdroplet Formation

Photographs of the droplets that formed at each oil phase flow rate (V_O_) are shown in Figure 4. In the experiment using PBS as the aqueous phase (Figure 4A), the droplet size decreased as the V_O_ increased. This may be due to the increase in shear force caused by the increase in the flow rate of the oil phase. In the experiment using PEG solution as the aqueous phase (Figure 4B), the droplet size also decreased with increasing V_O_, but the droplet size was larger than that of PBS.

Figure 5 shows the droplet size at each V_O_. The droplet diameter (D) of the aqueous phase with PBS was 232 ± 6 μm at 4 μL min^−1^, but D decreased with increasing V_O_ and converged to about 90 μm. The coefficient of variation of D was 8.6% at the V_O_ of 36 μL min^−1^, and 2.2~4.2% at other V_O_. In the experiment using PEG solution as the aqueous phase, D was 272 ± 10 μm at the V_O_ of 4 μL min^−1^. The D decreased with the increase in the V_O_ and converged to about 180 μm. The coefficient of variation of D was 1.9~3.6%. Since the coefficient of variation of D is less than 5%, it can be said that the droplets formed in this study are extremely monodisperse.

## 4. Discussion

In this study, we were able to fabricate microfluidic devices for the formation of microdroplets using a consumer-grade laser cutter. Compared to conventional microfluidic devices fabricated using photolithography, we were able to fabricate microfluidic devices using simple equipment and methods. In addition, we were able to fabricate microfluidic channels that are smaller in width (0.15 mm) than those in our previous studies (0.5 mm [19] and 1 mm [20]) by optimizing laser-cutting parameters. Therefore, we were able to expand the application of microfluidic devices fabricated using a consumer-grade laser cutter. The cross-sectional shape of the channel was trapezoidal because the laser beam was focused on the top surface of the substrate during the cutting process, and more acrylic was removed from the top surface than from the bottom surface. The plug flow and microdroplets were successfully observed, which indicates that the roughness of the channel ceiling is not a problem for observation.

The volume of the droplets formed in this study can be considered as follows. In the previous study [23], it was shown that the volume of the droplet converges to a constant value as the time required for shear force to tear off the droplet becomes shorter. The reason why the droplet size is larger in the experiment using PEG solution than in PBS is due to the difference in viscosity. The viscosity of PBS is 0.89 mPa s at 25 °C, which is assumed to be the same as the viscosity of water. On the other hand, the viscosity of 10% PEG (average MW: 7000–9000) solution was reported to be 8.9 mPa s at 25 °C [24]. Therefore, since the viscosity of the PEG solution is 10 times larger than that of the PBS, it takes a long time to tear off the PEG solution compared to PBS, resulted in the formation of larger droplets.

The droplet formation method developed in this study has many possibilities. First, droplet-based biomolecule analysis, such as ddPCR, can be realized with the simple device reported in this study. If cells are trapped in the droplet, it is possible to analyze intracellular biomolecules inside the droplets. Another direction is the application to the formation of artificial cells. The aqueous PEG solution used in this study is often used to mimic the molecular crowding in cells [25]. Therefore, if biomolecules are added to the solution and divided into droplets, it will be possible to evaluate them in an environment that mimics cells in terms of both solution composition and size. Through such efforts, we expect to make further progress in understanding biochemical processes in cells.

## Figures and Tables

**Figure 1 micromachines-12-00160-f001:**
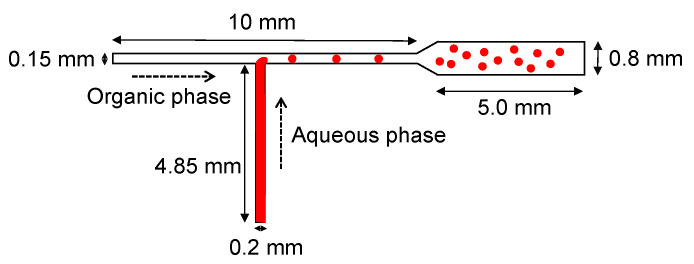
Schematic illustration of microdroplet formation in a T-shaped microfluidic channel.

**Figure 2 micromachines-12-00160-f002:**
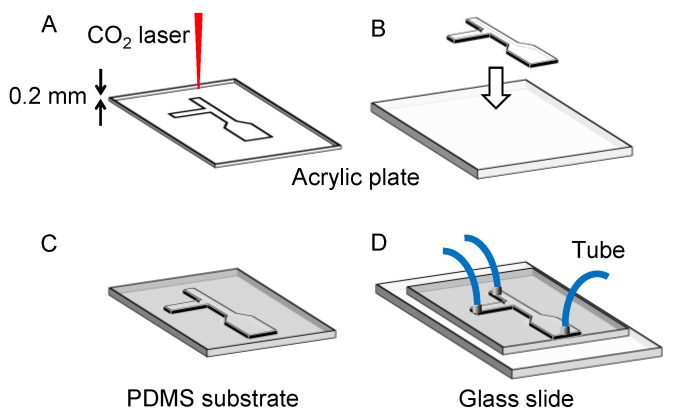
Schematic illustrations of fabrication procedures of the microfluidic device. (**A**) Cutting acrylic plate with CO_2_ laser. (**B**) Bonding. (**C**) Molding. (**D**) Bonding and tubing.

**Figure 3 micromachines-12-00160-f003:**
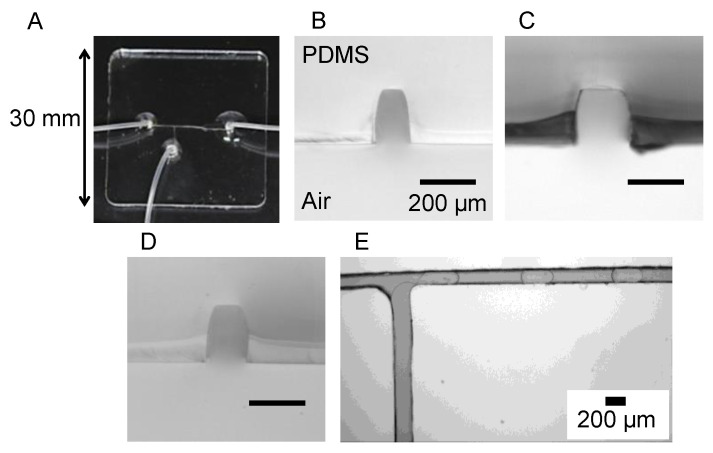
(**A**) A picture of a microfluidic device. (**B**–**D**) Microscopic images of cross section of recessed microchannel patterns on PDMS substrate. Images were taken at (**B**) left part, (**C**) bottom part, and (**D**) right part of the T-junction. Scale bars: 200 μm. (**E**) A microscopic image of plug flow in a T-shaped microchannel. Dispersed phase: PBS (1 μL min^−1^). Continuous phase: Oil for droplet digital PCR (4 μL min^−1^).

**Figure 4 micromachines-12-00160-f004:**
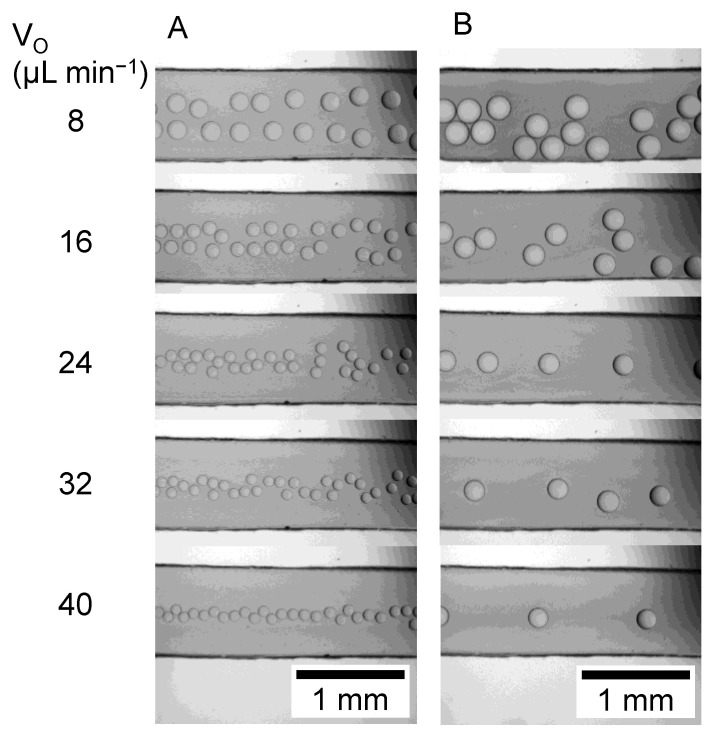
Microscopic images microdroplets observed in a T-shaped microchannel. V_O_ denotes the volume flow rate of continuous phase. The volume flow rate of the dispersed phase was fixed to 1 μL min^−1^. Dispersed phase: (**A**) PBS, (**B**) 10% PEG solution. Continuous phase: Oil for droplet digital PCR.

**Figure 5 micromachines-12-00160-f005:**
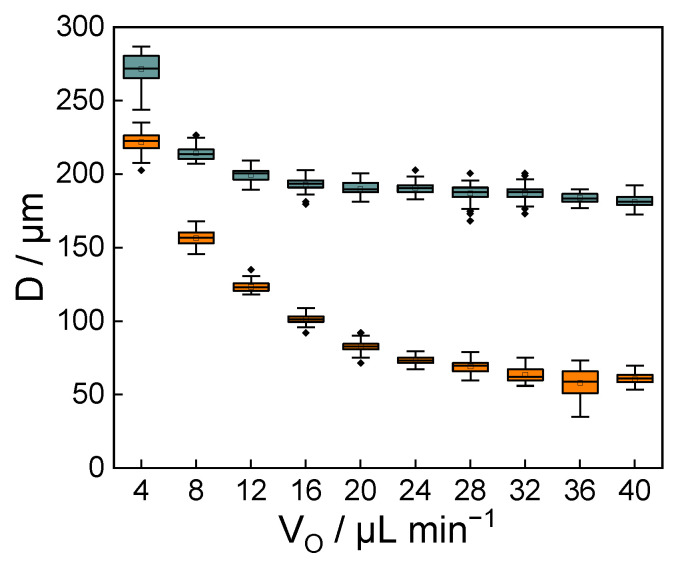
A box plot showing the dependence of the diameter of microdroplets on V_O_. The volume flow rate of the dispersed phase was fixed to 1 μL min^−1^. Dispersed phase: (Orange) PBS, (Blue) 10 % PEG solution. N = 50.

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
