# Peer review of "Fabrication of a T-Shaped Microfluidic Channel Using a Consumer Laser Cutter and Application to Monodisperse Microdroplet Formation"

_micromachines, 2021, doi:10.3390/mi12020160_

Round 1

Reviewer 1 Report

The authors proposed a new method using laser cutter to fabricate a microchannel and tested its feasibility by applying to droplet formation. The manuscript is concise and correct length. The generated microdroplets looks monodisperse and this method technically could be simpler compared to photolithogrphy. However, this manuscript somewhat limits the novelty of the current methodology and scientific improvement, except for simplicity. The characterization of this methods need to be more extensive for the paper publication. The following revisions are suggested.

  1. Reference 15 and16 show similarity in terms of laser cutting technology, which somewhat limits the novelty of the current manuscript.
  2. The author need to show any efficiency improvement or any advantages in terms of droplet generation compared to the microfluidic chip fabricated by conventional photolithography.
  3. In bonding method, is the glue the author used not biologically toxic? Does this glue have no effect on toxicity during droplet generation used for cells or biological samples?
  4. Also, it seems that this glue may cause an uneven thickness differences across the entire device
  5. T-junction intersection doesn’t look sharp. Is this weakness of laser cutting?

Reviewer 2 Report

The manuscript presents a study on the laser fabrication of a T-Shaped Microfluidic Channel  for application in droplet Formation.

The fabrication of microfluidic device is not a new topic, but a new point of view can always be interesting. The novety of the paper (fabrication method, application??) should be better highlithed, to give more prominance to the results.

Some issues should be adressed.

INTRO

Line 46 : add a ref

Line 53 add refs

Lines 56-57: Do you not need specialized software such as CAD for your research?

Lines 54-64: In these lines, the authors would like to provide an overview of te main techniques used for microfluidic devices. However, soft litography and laser microfabrication are not adeguatelly described, despite their spread in the recent years. Some interesting publication about these topics are:

https://doi.org/10.1007/s10404-019-2206-1

doi:10.3390/inventions3030060

doi: 10.1007/s10404-019-2206-1

What are the main advantages/difference of your laser method respect to others laser-based fabrication platform? What are the novelty of your reserch?  These points should be discussed.

RESULTS and DISCUSSION

In the title and keywords, the authors stress the laser fabrication of the device. So, in the 3.1 section, some results about fabrication should be presented; such as, shape/section of the channel (3D microscopic images) , roughness of the bottom part of the channel, dimensions...etc.

line 143: what does small mean? 100 um, 1 mm?

Round 2

Reviewer 1 Report

Authors clarified the significant points rasied by reviewers

Reviewer 2 Report

The authors  have improved the manuscipt according to the referees comments.